

**Revised treatment of wet scavenging processes dramatically improves GEOS-Chem**
**12.0.0 simulations of nitric acid, nitrate, and ammonium over the United States**
Gan Luo, Fangqun Yu, and James Schwab
Atmospheric Sciences Research Center, University at Albany
**Abstract**
The widely used community model GEOS-Chem 12.0.0 and previous versions have
been recognized to significantly overestimate the concentrations of gaseous nitric acid,
aerosol nitrate, and aerosol ammonium over the United States. The concentrations of
nitric acid are also significantly over-predicted in most global models participating a
recent model inter-comparison study. In this study, we show that most or all of this
overestimation issue appears to be associated with wet scavenging processes.
Replacement of constant in-cloud condensation water (ICCW) assumed in GEOS-Chem
standard versions with one varying with location and time from the assimilated
meteorology significantly reduces mass loadings of nitrate and ammonium during the
wintertime, while the employment of an empirical washout rate for nitric acid
significantly decreases mass concentrations of nitric acid and ammonium during the
summertime. Compared to the standard version, GEOS-Chem with updated ICCW and
washout rate significantly reduces the simulated annual mean mass concentrations of
nitric acid, nitrate, and ammonium at surface mentoring network sites in US, from 2.04 to
1.03 $\mu g\ m^{-3}$, 1.89 to 0.88 $\mu g\ m^{-3}$, 1.09 to 0.68 $\mu g\ m^{-3}$, respectively, in much better
agreement with corresponding observed values of 0.83, 0.70, and 0.60 $\mu g\ m^{-3}$,
respectively. In addition, the agreement of model simulated seasonal variations of
corresponding species with measurements is also improved. The updated wet scavenging
scheme improves the skill of the model in predicting nitric acid, nitrate, and ammonium
concentrations which are important species for air quality and climate.

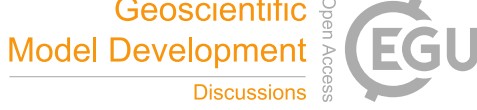

## 1. Introduction

Nitrate and ammonium are important secondary inorganic aerosols in the atmosphere, contributing significantly to total aerosol mass over most polluted regions (Bian et al., 2017) and to aerosol direct radiative forcing over urban and agriculture regions (Bauer et al., 2007; Myhre et al., 2013). The amount of nitrate and ammonium also regulates the concentration of gaseous ammonia which often plays an important role in the formation of new particles (Kirkby et al., 2011; Yu et al., 2018). In addition, nitrate and ammonium help newly formed particles grow to larger sizes suitable for cloud condensation nuclei (Yu and Luo, 2009) and thus can impact aerosol indirect radiative forcing (Twomey, 1977).

Nitric acid, nitrate, and ammonium concentrations are often overestimated by atmospheric models (Pye et al., 2009; Walker et al., 2012; Bian et al., 2017; Zakoura and Pandis, 2018), including the widely used community model GEOS-Chem (e.g., Zhang et al., 2012; Heald et al., 2012). Zhang et al. (2012) studied nitrogen deposition over the US with GEOS-Chem and found both nitric acid and nitrate concentrations are overestimated, especially in wintertime. They suggested that this is the result of excessive nitric acid formation via night time chemistry of heterogeneous $N_2O_5$ hydrolysis. However, Heald et al. (2012) found the overestimate of heterogeneous $N_2O_5$ hydrolysis does not fully account for the nitrate bias and suggested the positive nitrate bias is likely linked with an overestimate of nitric acid concentrations. Heald et al. (2012) investigated other possible causes for the overestimation of nitric acid concentrations arising from uncertainties in daytime formation and dry deposition, and concluded that none of these uncertainties could fully account for the reduction in nitric acid required to correct the nitrate bias. Based on comparisons of simulated nitrate and ammonium aerosol from nine AEROCOM models with ground station and aircraft measurements, Bian et al. (2017) concluded that most models overestimate surface nitric acid volume mixing ratio by a factor of up to 3.9 over North America and the overestimation cannot be simply attributed





to model uncertainties. Backes et al. (2016) suggested that uncertainties in the temporal
profiles of ammonia emissions could also contribute significantly to the bias of nitrate
concentrations. However, the impact of ammonia mostly happened during summer time.
Zakoura and Pandis (2018) found significant decrease in nitrate concentration when they
enhanced their model resolution from 36 km × 36 km to 4 km × 4 km in the PMCAMx
model. However, similar results are not found in global models with much coarser grids
than regional models. All these studies indicate that the overestimation of nitric acid,
nitrate, and ammonium mass concentrations in current atmospheric chemistry models
remains to be resolved.

10        In this study, we proposed an improved treatment of wet scavenging in GEOS-Chem

by considering cloud condensation water variability and empirical washout rate, which
together significantly improve the estimates of nitric acid, nitrate, and ammonium over
the US. The improved wet scavenging in GEOS-Chem is described in section 2. The
comparison of model results with in-site observations and the changes of the three
species over the US are presented in section 3. Section 4 is the summary and discussion.
**2. Improved scheme for wet scavenging**

18        Wet scavenging is the main removal pathway for many atmospheric air pollutants.

Two mechanisms are involved in wet scavenging: rainout (in-cloud scavenging) and
washout (below-cloud scavenging). GEOS-Chem treats wet scavenging associated with
stratiform and convective precipitation separately.

**2.1 Impact of in cloud condensed water (ICCW)**

24        For stratiform precipitation, in the most recently released GEOS-Chem version

12.0.0 (GC12), rainout is parameterized according to Jacob et al. (2000) as

26        $$F = \frac{P_r}{k \cdot ICCW}\left(1 - e^{-k \cdot \Delta t}\right) \quad (1)$$

where $F$ is the fraction of a soluble tracer in the grid-box scavenged by rainout, $\Delta t$ is the





model integration time step. $k$ is the first-order rainout loss rate which represents the
conversion of cloud water to precipitation water. $ICCW$ represents the condensed water
content (liquid) within the precipitating cloud (i.e., in cloud) and $P_r$ is the rate of new
precipitation formation in the corresponding grid-box.

5        The rainout loss rate ($k$) represents how fast cloud condensation water can be

removed from the atmosphere and thus is critical for rainout scavenging. $k$ is defined in
Jacob et al. (2000) and coded in GC12 (called $k_{GC12}$ thereafter) as
$$k_{GC12} = k_{min} + \frac{P_r}{ICCW} \quad (2)$$
where $k_{min}$ is the minimum value of rainout loss rate derived from the stochastic
collection equation which indicates that in one hour at least ~ 0.36 of cloud droplets are
lost to autoconversion/accretion (Beheng and Doms 1986). In GC12, $k_{min}$ is set to be 0.36
$hr^{-1} = 1 \times 10^{-4}$ $s^{-1}$.

13       It should be noted that $P_r$ in Eq. (2) is a grid-box mean value, while $ICCW$ is an in

cloud value. To be physically consistent, we suggest a new expression of $k$ ($k_{new}$) that
replaces grid-box mean $P_r$ with the corresponding in cloud value $P_r/f_c$.
$$k_{new} = k_{min} + \frac{P_r}{f_c \cdot ICCW} \quad (3)$$
where $f_c$ is the grid-box mean cloud fraction. As we will show later, Eq. (3) gives $k$ values
in much better agreement with those derived from cloud model simulations and
observations.

20       To calculate $F$, GC12 uses $P_r$ from the Modern-Era Retrospective analysis for

Research and Applications Version 2 (MERRA2) meteorological fields. For $ICCW$ in Eqs.
1-3, Jacob et al. (2000) used a constant value of 1.5 g m$^{-3}$ and Wang et al. (2011) changed
it to 1 g m$^{-3}$. In GC12, the default value of $ICCW$ is 1 g m$^{-3}$. However, $ICCW$ in the
atmosphere varies with time and location. Here we suggest to use time and location
dependent $ICCW$ (named $ICCW_t$) which can be derived from MERRA2 meteorological
fields as





$$ICCW_t = \frac{CW + P_r \cdot \Delta t}{f_c} \quad (4)$$
where $CW$ is grid-box mean cloud water content, while $P_r \cdot \Delta t$ represents rain water
content produced during the time step $\Delta t$.
Figure 1a shows seasonal variations of $ICCW_t$ (Eq. 4) averaged throughout the lower
troposphere (0–3 km) of the whole globe ($ICCW_{t\_G}$), over all land surface ($ICCW_{t\_L}$),
over the oceans ($ICCW_{t\_O}$), and over the continental US ($ICCW_{t\_US}$). For comparisons, the
constant values of $ICCW$ assumed in Jacob et al. (2000) ($ICCW_{J2000}$) and GC12
($ICCW_{GC12}$) are also shown. The monthly mean values of $ICCW_{t\_G}$, $ICCW_{t\_L}$, $ICCW_{t\_O}$,
and $ICCW_{t\_US}$ vary within the ranges of 0.90–1.03 g m$^{-3}$, 0.30–0.45 g m$^{-3}$, 1.15–1.26 g
m$^{-3}$, and 0.21–0.53 g m$^{-3}$, respectively. This figure shows that $ICCW_{t\_G}$ is close to the
assumed $ICCW$ value of 1 g m$^{-3}$ used in GC12. As can be seen from Fig.1a, $ICCW_{t\_O}$ is
greater than 1 g m$^{-3}$, but $ICCW_{t\_L}$ is much less than the constant value of 1 g m$^{-3}$ assumed
in GC12. The mean $ICCW$ over the continental US (bright green line) is close to $ICCW_{t\_L}$
(olive line), and is $\sim 5$ times less than the assumed value in GC12 during the wintertime
and $\sim 2$ times less during the summertime. As we will show later, the constant $ICCW$ of 1
g m$^{-3}$ assumed in GC12 leads to significant underestimation of rainout over the
continental US, especially during the wintertime.
Figure 1b shows seasonal variations of mean $k_{GC12}$, $k_{new}$, and $k_{new\_ICCWt}$ in the lower
troposphere (0-3 km) of the continental US. Referring to Eq. (2), the figure shows that
$k_{GC12}$ is dominated by $k_{min}$ (which is physically unsound) and thus shows negligible
seasonal variation. Conversely, $k_{new}$ is low in the wintertime and high in the
summertime. $k_{new\_ICCWt}$ is 2.3 times higher than $k_{new}$ during January and 1.6 times higher
than $k_{new}$ during July. Both $k_{new}$ and $k_{new\_ICCWt}$ are within the range of rainout loss rates
($10^{-4}$–$10^{-3}$ s$^{-1}$) indicated by cloud model simulations and estimates based on observations
(Giorgi and Chameides, 1986).



1       From Eqs. (1), (3), and (4), we can get the updated parameterization for rainout loss

fraction at each location and time step

$$F = \frac{f_c \cdot P_r}{k_{new\_ICCW_t}(CW + P_r \cdot \Delta t)}\left(1 - e^{-k_{new\_ICCW_t} \cdot \Delta t}\right) \quad (5)$$

**2.2 Impact of empirical washout rate on nitric acid wet scavenging**

6       Still considering the case of stratiform precipitation in GOES-Chem, the fraction of

aerosols and $HNO_3$ within a grid-box that is scavenged by washout over a time step is
parameterized as (Wang et al., 2011; Liu et al., 2001; Jacob et al., 2000)

9       $F_{wash} = f_r(1 - exp(-k_{wash}\Delta t))$   (6)

10       $f_r = \max(\frac{P_r}{k \cdot ICCW}, f_{top})$   (7)

11       $k_{wash} = \Lambda \left(\frac{P_r}{f_r}\right)^b$   (8)

where $f_r$ is the horizontal areal fraction of the grid-box experiencing precipitation and $f_{top}$
is the value of $f_r$ in the layer overhead ($f_{top} = 0$ at the top of the precipitating column).
$k_{wash}$ is washout rate, $\Lambda$ is washout scavenging coefficient, and $b$ is an exponential
coefficient. In the original GEOS-Chem, $\Lambda = 1$ cm$^{-1}$ and $b = 1$ for both aerosols and nitric
acid (Liu et al., 2001; Jacob et al., 2000).

17       It has been well recognized that, for aerosols, $\Lambda$ and $b$ depend on particle size (Wang

et al., 2010; Feng, 2007; Andronache et al., 2006; Henzing et al., 2006; Laakso et al.,
2003). Feng (2007) suggested values of $b = 0.62$, 0.61, and 0.8 for particles in nucleation
(diameter 1 nm – 40 nm), accumulation (40 nm – 2.5 μm), and coarse mode (>2.5 μm),
respectively. Many studies indicate that there are large difference between existing
theoretical and observed size-resolved washout rates (Wang et al., 2010; Andronache et
al., 2006; Henzing et al., 2006; Laakso et al., 2003). For particles within the diameter
range of 0.01–2 μm, size-resolved washout rates derived from analytical formulas are one
to two orders of magnitude smaller than those derived from field measurements (e.g.,



Wang et al., 2010). This large difference could result from turbulent flow fluctuations
(Andronache et al. 2006; Khain and Pinsky, 1997), vertical diffusion process (Zhang et al.,
2004), and droplet-particle collection mechanisms (Park et al., 2005).

4       In GC12, $\Lambda$ and $b$ for aerosols are parameterized as a function of particle size modes

(Wang et al., 2011), following Feng (2007). For nitric acid, GC12 keeps $\Lambda$ = 1 cm$^{-1}$ and $b$
= 1, unchanged from the original CEOS-Chem parameters. In this study, we employ the
size-dependent aerosol washout parameterization derived from six years of field
measurements over forests in southern Finland (Laakso et al., 2003; Wang et al., 2010).
We further estimate nitric acid washout scavenging coefficients by referring to field
measurements for particles of 10 nm (Laakso et al., 2003) and the theoretical dependence
of scavenging coefficients on particle sizes for particles < 10 nm (Henzing et al., 2006).
The collection efficiency of particles smaller than 10 nm by rain droplets is dominated by
Brownian diffusion, and in this regard we can treat nitric acid as a single molecule (or
particle) with diameter of 0.5 nm. Through this approach, we derive an empirical $\Lambda$ value
for nitric acid of 2 cm$^{-1}$. In addition, we adopt the $b$ value of 0.62 for nucleation mode
particles (diameter 1 nm – 40 nm) (Feng, 2007) for nitric acid. When in cloud
precipitation intensity is 1 mm h$^{-1}$, this empirical washout loss rate equals $3\times10^{-3}$ s$^{-1}$
which is about two orders of magnitude larger than the corresponding washout loss rate
(0.1 hr$^{-1}$ = $2.8\times10^{-5}$ s$^{-1}$) currently in GC12.
For convective precipitation, MERRA2 meteorological fields do not provide
convective cloud fraction and water content. Therefore, the updated wet scavenging
method discussed above for stratiform precipitation cannot be directly applied to
convective precipitation rainout scavenging in GEOS-Chem. However, the empirical
value for nitric acid washout is also applied to convective washout in the present study as
Case 4.

**3. Model simulations and results**

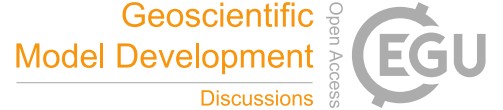

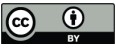

1       To study the impacts of various updates to the wet scavenging as described in

Section 2 on model simulated nitric acid, nitrate, and ammonium mass concentrations, we
run GEOS-Chem for 4 cases: (1) standard GC12 parameterizations for rainout and
washout (Keller et al., 2014; Fontoukis and Nenes, 2007; Martin et al., 2003; Bey et al.,
2001), called GC12; (2) same as the Case GC12 except $k_{new}$ in Eq. 3 is used, called Knew;
(3) same as the Case Knew except $ICCW_t$ from MERRA2 (Eq. 4) is used, called $ICCW_t$;
(4) same as the Case $ICCW_t$ except empirical washout rates for nitric acid and aerosols
are used, called $ICCW_t\_EW$. For each case, we carry out simulations from December
2010 to December 2011, with the first month as spin-up. The model horizontal resolution
is $2^o \times 2.5^o$ and vertically there are 47 layers. The present analysis focuses on the
continental United States. We compared simulated nitric acid with in-situ observations at
Clean Air Status and Trends Network (CASTNET) sites, simulated nitrate and
ammonium with in-situ observations at Interagency Monitoring of Protected Visual
Environments (IMPROVE) and Chemical Speciation Network (CSN) sites. For 2011,
there were 74 sites with available nitric acid observations from CASTNET. For the same
year, IMPROVE had 120 sites with available nitrate and ammonium observations, while
CSN had 94 sites with available nitrate observations and 63 sites with available
ammonium observations.

19       The effects of different modifications to the GC12 wet scavenging parameterization

on model simulated nitric acid, nitrate, and ammonium mass concentrations are shown in
Figures 2-3 and Table 1. Most of the changes of mass concentrations of the 3 species over
the US are caused by the changes of cloud condensation variability and/or empirical
washout rate. The impact of new rainout loss rate ($k_{new}$) is relatively small because of the
cancelling effect of $k$ in the denominator and also in the exponent in Eq. 1. As shown in
Figs. 2a-2b and Table 1, all cases except $ICCW_t\_EW$ overestimate nitric acid at
CASTNET sites by a factor 2–3 in both wintertime and summertime. Consideration of
cloud condensation water variability slightly reduces nitric acid in January and December

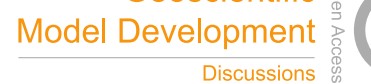

but has negligible effect during other months. The inclusion of the empirical washout rate
reduces the normalized mean bias (NMB) of nitric acid from ~150 % to 24 % (Table 1).
Figures 2c and 2d show the impacts of improved wet scavenging on nitrate. It is clear that
GC12 significantly overestimates nitrate concentration at most sites especially during the
wintertime, in agreement with previous studies (Heald et al., 2012; Walker et al., 2012).
Replacing constant ICCW with variable $ICCW_t$ reduces the NMB of nitrate from 170 %
to 84 %. ICCW has significant impact on reducing nitrate mass concentration during the
wintertime and a smaller impact during the summertime. Wintertime bias of nitrate was
reduced from 2 μg m$^{-3}$ to 0.7 μg m$^{-3}$. The change of washout rate from theoretical value
to empirical formula results in an additional 59 % reduction of NMB for nitrate and
impacts nitrate mass concentration significantly both in the winter and in the summer. For
ammonium, NMB is reduced from 85 % to 43 % after considering rainout with variable
cloud condensation water. Similar to nitrate, the impact of CCW is large during the
wintertime and smaller during the summer time. After considering empirical washout, the
NMB of ammonium is reduced to 13 %. While the update in the wet scavenging
parameterization significantly improves agreement of the model simulated mass
concentrations nitric acid, nitrate, and ammonium over the US with those observed, it
does not affect the correlation coefficients of annual mean values (Table 1) which are
dominated by spatial distributions (Fig. 3).
Figure 3 shown the horizontal distributions of surface layer nitric acid, nitrate, and
ammonium mass concentrations over the US for case GC12 (a-c) and case $ICCW_t$_EW
(d-f). For comparison, annual mean mass concentrations observed at CASTNET,
IMPROVE, and CSN sites are shown in filled cycles. The spatial pattern of the simulated
concentrations of the three species for the ICCW_EW case is close to those for the GC12
case. High concentrations of nitric acid are mainly located at northeastern, southern, and
western US with the values up to 2–4 μg m$^{-3}$ based on GC12 (Fig. 3a) and 1-2 μg m$^{-3}$
based on $ICCW_t$_EW (Fig. 3d). Horizontal distribution of nitrate is different from that of





nitric acid. Nitrate is mainly located at the Ohio valley region and the Northeastern US
with values up to 4–5 μg m$^{-3}$ based on GC12 (Fig. 3b) and 1-3 μg m$^{-3}$ based on
ICCW$_t$_EW (Fig. 3e). Ammonium shows a similar horizontal distribution to that of
nitrate, but its value is ~50 % lower than nitrate concentration. For the whole continental
US domain, the annual mean nitric acid, nitrate, and ammonium concentration in the
model surface layer are reduced from 1.48 μg m$^{-3}$ to 0.78 μg m$^{-3}$, 1.03 μg m$^{-3}$ to 0.46 μg
m$^{-3}$, 0.76 μg m$^{-3}$ to 0.47 μg m$^{-3}$, respectively. The percentage changes for nitric acid,
nitrate, and ammonium concentrations averaged within the domain are -47%, -55%, and
-38%, respectively. The improved wet scavenging treatment had significant impacts on
nitric acid, nitrate, and ammonium modeling over the US. As can be seen from Figs.
3a-3f (and also Fig. 2 and Table 2), simulated nitric acid, nitrate, and ammonium mass
concentrations over the US based on the updated wet scavenging parameterization (i.e.,
ICCW$_t$_EW) are in much better agreement with in-situ measurements.
**4. Summary and discussions**
We present an improved wet scavenging parameterization for use in in GEOS-Chem
by considering cloud condensation water variability and an empirical washout rate. The
updated parameterization significantly reduces the overestimation of simulated annual
mean mass concentrations of nitric acid, nitrate, and ammonium at CASTNET,
IMPROVE, and CSN sites in US, from 2.04 to 1.03 (observation: 0.83) μg m$^{-3}$, 1.89 to
0.88 (observation: 0.70) μg m$^{-3}$, 1.09 to 0.68 (observation: 0.60) μg m$^{-3}$, respectively. In
addition, the agreement of model simulated seasonal variations of corresponding species
with measurements is also improved. The updated wet scavenging scheme provides a
partial solution to the persistent problem of nitric acid and nitrate overestimation in the
widely used community model GEOS-Chem (e.g., Heald et al., 2012) and thus improve
the skill of the model in predicting nitric acid, nitrate, and ammonium concentrations.
The empirical washout rate suggested in the present work will also help to resolve the





significant over-prediction of nitric acid by most of the 9 global models participating in the Aerosol Comparisons between Observations and Models (AeroCom) phase III study (Bian et al., 2017). Due to large difference in nitric acid washout rate based on theoretical and field studies and the importance of this rate, further research is needed to better understand the underlying reasons and reduce the difference. At the time being, we recommend the empirical values to be used in models.

While the present study focused on the US where abundant relevant measurements are available, the updated wet scavenging parameterization impacts model simulated nitric acid, nitrate and ammonium concentrations in other regions as well, particularly over land. The changes of nitrate and ammonium mass concentrations not only impact particle growth but also influence ammonia concentrations which are important for aerosol nucleation (Kirkby et al., 2011; Yu et al., 2018), via the equilibrium of sulfate-nitrate-ammonium. The updated scheme presented in this study has potential implications to new particle formation, particle growth, aerosol size, CCN number concentration and associated radiative forcing, which will be the subjects of future research.

Code and data availability. The code of GEOS-Chem 12.0.0 is available through the GEOS-Chem distribution web-page http://wiki.seas.harvard.edu/geos-chem/index.php/GEOS-Chem_12. All measurement data are publicly available.

Acknowledgments. This work is supported by NYSERDA under contract 100416, NASA under grant NNX13AK20G, and NSF under grant 1550816. We would like to acknowledge Interagency Monitoring of Protected Visual Environments (IMPROVE), Chemical Speciation Network (CSN), and Clean Air Status and Trends Network (CASTNET) for the in-site measurement data. GEOS-Chem is a community model


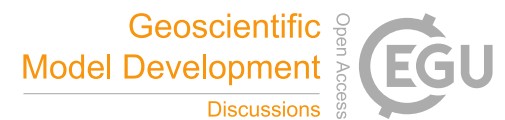

maintained by the GEOS-Chem Support Team at Harvard University.

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



Table 1. Observed annual mean surface concentrations of $HNO_3$, nitrate, and ammonium
at CASTNET, IMPROVE, and CSN sites. Annual mean surface concentrations (Mean),
normalized mean bias (NMB), and correlation coefficient ($r$) between observed and
simulated annual mean values for the 3 species by GC12, Knew, $ICCW_t$, and $ICCW_t$_EW
cases.

| | $HNO_3$ | | | NIT | | | NH4 | | |
|---|---|---|---|---|---|---|---|---|---|
| | Mean ($\mu g\ m^{-3}$) | NMB (%) | $r$ | Mean ($\mu g\ m^{-3}$) | NMB (%) | $r$ | Mean ($\mu g\ m^{-3}$) | NMB (%) | $r$ |
| Observation | 0.83 | | | 0.70 | | | 0.60 | | |
| GC12 | 2.04 | 145.1 | 0.73 | 1.89 | 168.1 | 0.53 | 1.09 | 81.4 | 0.75 |
| Knew | 2.05 | 146.8 | 0.73 | 1.90 | 170.5 | 0.53 | 1.11 | 84.5 | 0.75 |
| $ICCW_t$ | 1.87 | 125.0 | 0.74 | 1.29 | 83.5 | 0.57 | 0.86 | 42.7 | 0.78 |
| $ICCW_t$_EW | 1.03 | 24.2 | 0.72 | 0.88 | 25.0 | 0.57 | 0.68 | 12.8 | 0.78 |


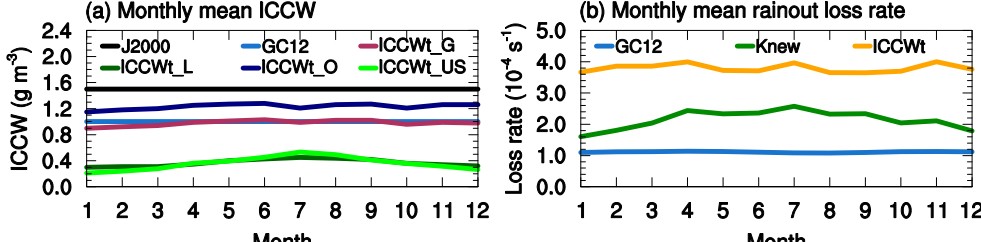

Figure 1. (a) Monthly variations of ICCW averaged over the lower troposphere layers of
the whole globe (maroon), global land areas (olive), global oceans (navy), and
continental US (green) from MERRA2, along with constant ICCW values assumed in
J2000 (black) and GC12 (blue). (b) Monthly variations of the rainout loss rate averaged
in the lower troposphere layers of the continental US based on Eq. (2) (i.e, GC12) and Eq.
(3) with constant ICCW of 1 g m$^{-3}$, and Eq. (3) with MERRA2 ICCW (Eq. 4).



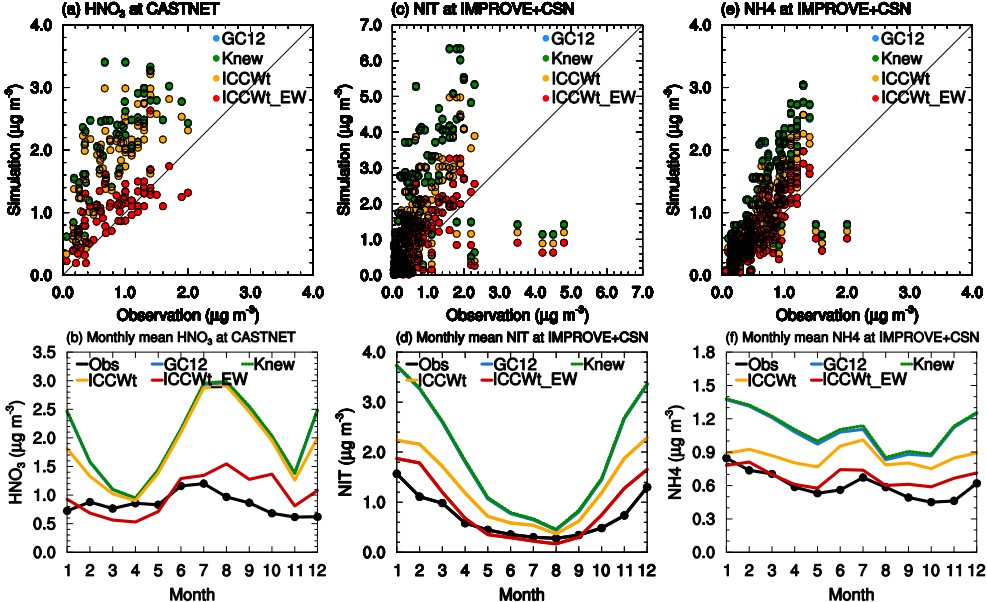

Figure 2. (a) Scatter plot of observed and simulated annual mean HNO$_3$ at CASTNET
sites and (b) monthly variations of median showing the comparison between nitric acid
mass concentrations observed at CASTNET sites (black) and simulated by GC12 (blue),
Knew (green dash), ICCW$_t$ (yellow), and ICCW$_t$_EW (red) cases. (c) and (d) are the
same as (a) and (b) but for nitrate at IMPROVE+CSN sites. (e) and (f) are the same as (a)
and (b) but for ammonium at IMPROVE+CSN sites.





Figure 3. Horizontal distributions of surface layer nitric acid, nitrate, and ammonium simulated by the GC12 case (a-c) and the ICCW$_t$_EW case (d-f). Filled circles are annual mean mass concentrations observed at CASTNET, IMPROVE, and CSN for corresponding species.