# Peer review of "Revised treatment of wet scavenging processes dramatically improves GEOS-Chem"

_Geoscientific Model Development, 2019_

## Referee Comment (RC1) · Anonymous Referee #1 · 8 May 2019

The authors aim to improve wet deposition simulation of nitric acid, nitrate and ammonium over the United States using GEOS-Chem via updating both in cloud stratiform cloud scavenging and below cloud washout. For in cloud scavenging, they adopt the dynamic varied condensed water content provided by MERRA2 meteorological fields, needed in wet scavenging parameterization, instead of the current assumption of a global flat value. For below cloud washout, they derive a new set of empirical washout scavenging coefficient and exponential coefficient for nitric acid based on the size-resolved coefficients summarizing from field measurement and theoretical derivation.

[Figure]

This is an interesting and valuable study. The study would have a potential impact on the broad atmospheric composition study via improving tracers' wet scavenging if the authors could validate their work for other aerosols and their precursors. A minor revision is required before the paper is published in GMD.

Major Comments

The authors test the physical-based condensed cloud water for stratiform cloud rainout. Convective cloud removal is important and is necessary to be studied as well. Studying convective rainout is particularly important for using the current generation of NASA GEOS meteorological fields since its partitioning of large scale and convective clouds tilts more towards the latter. The convective cloud fraction and water content can be provided by the GEOS model.

The authors are highly encouraged to evaluate and summarize the impact of their work on other aerosols and their precursors. Once the GEOS-Chem adopts the improvements in wet scavenging parameterization suggested by the authors, all aerosols and their precursors undergoing wet scavenging will be impacted. To have confidence in using their work, they should at least provide a brief description of the model performance for all important aerosol fields in supplementary material. In addition, the authors' work focuses on the United States only. What is the anticipated influence of the improved wet scavenging on other regions?

To be more useful of the proposed work on wet deposition, more words are needed about the broader impact of the study on the whole atmospheric chemistry community. Can other global chemistry models adopt their improvement? Is there anything that other modelers should be cautioned of in adopting their work?

Specific comments

1. Page 3 line 26 equation 1: Should the k in exponential term differ with the k in the denominator of coefficient? For my understanding, the k in exponential term, which is

the first-order rainout rate, is linked to specific tracer species. One the other hand, the k in denominator represents the generic conversion rate of cloud water to precipitation. Please double check this. Please also give units of these fields and parameters in equation 1. 2. Page 7 lines 4-19: How about aerosols? Should the washout scavenging coefficients of aerosols be adjusted accordingly? 3. Page 8 line 7: Please add one more case study. Similar to case study 4 but empirical washout rate of HNO3 is applied only to large scale precipitation. This case, combined with case 4, will give us further information about the relative washout contribution from large scale and convective scale precipitations. 4. Page 8 line 7: Do the authors present the work of empirical washout rates for aerosols? Section 2.2 seems only give discuss for HNO3. What are the new empirical washout rates for aerosols? 5. Page 9 lines 1-2: The change range shown here (from 150% to 24%) includes not only using empirical washout rate, but also changing cloud condensation water.

---

## Referee Comment (RC2) · Anonymous Referee #2 · 19 May 2019

This paper presented a revised wet scavenging parameterization that considers the spatiotemporal variability of cloud liquid water content and an empirical washout (below-cloud scavenging) rate in the GEOS-Chem global chemical transport model. The authors showed that the updated parameterization significantly improves simulated annual mean (and seasonal) mass concentrations of nitric acid, nitrate, and ammonium as compared with surface observations over the U.S. This is an important contribution to the improvement of GEOS-Chem. Minor revision is recommended before publication on GMD.

[Figure]

Major comments:

The impact of updated wet scavenging on model simulations was only assessed at the surface level and for nitric acid, nitrate, and ammonium over the U.S. It's not shown how the updated treatment of scavenging affects the global aerosol simulations, especially the vertical profiles and other aerosol species (e.g., sulfate). Consider discussing this in the Summary and Discussions section. Lead-210 aerosol tracer has been used to test wet deposition in GEOS-Chem (e.g., Liu et al., 2001), and this updated scavenging parameterization will need to be tested with (at least) lead-210 before it is incorporated into the standard version of the model.

Page 5, equation 4: 1). "CW is grid-box mean cloud water content". What's the corresponding variable name in MERRA-2? Does it include both cloud liquid (QL) and ice (QI), or QL only? 2). It's not clear why the rain water term "Pr*DeltaT" is needed. There is no prognostic precipitation (no raining condensate) in MERRA-2 or GEOS-5. Prognostic cloud liquid and ice are autoconverted to estimate precipitation. Are "CW" values for pre-conversion or post-conversion? More explanation as well as references are needed.

Page 7, lines 20-23: The first-order rainout parameterization is not used for convective precipitation scavenging in GEOS-Chem driven by MERRA-2. Instead, scavenging in convective updrafts are coupled with convective transport (e.g., see section 2.3.1 of Liu et al., 2001).

Minor comments:

Title: Suggest adding "surface" to the title since this study examined the impact of revised scavenging on surface aerosol concentrations only.

Page 1, line 21: typo "mentoring" ("monitoring")

Page 3, lines 6-7: are there references for this statement?

Page 3, line 14: change "in-site observations" to "surface observations"
Page 3: A brief description of the GEOS-Chem model is needed here before discussing the wet scavenging scheme.

Page 3, section 2: See this webpage http://acmg.seas.harvard.edu/geos/geos_chem_narrative.html for "Narrative description (and how to cite GEOS-Chem)", which provides guidance on citing relevant model components. "The wet deposition scheme in GEOS-Chem is described by Liu et al. [2001] for water-soluble aerosols and by Amos et al. [2012] for gases. Scavenging of aerosol by snow and cold/mixed precipitation is described by Wang et al. [2011, 2014]." Suggest citing Jacob et al. (2000) along with one of these publications, where appropriate, since it is an unpublished document. The first-order rainout parameterization (equations 1 and 2) is based on Giorgi and Chameides (1986), which also needs to be referenced.

Specify the units for variables in all equations in the text.

Page 4, line 3: condensed water content includes liquid and ice phases. Do you revise warm cloud scavenging only here? Does "Pr" (rate of new precipitation formation) include snow? How about ice cloud scavenging?

Page 4, lines 23-25: Croft et al. (2016) previously used GEOS-5 cloud liquid and ice water content to replace the fixed value in their GEOS-Chem-TOMAS simulations. Consider citing that work here. (Croft, B., Martin, R. V., Leaitch, W. R., Tunved, P., Breider, T. J., D'Andrea, S. D., and Pierce, J. R.: Processes controlling the annual cycle of Arctic aerosol number and size distributions, Atmos. Chem. Phys., 16, 3665-3682, https://doi.org/10.5194/acp-16-3665-2016, 2016.)

Page 8, line 4: these references are not for rainout and washout parameterizations, but for the standard GEOS-Chem model (or other model components).

Page 9, line 13: CCW or ICCW?

Page 9, line 17: concentrations OF; line 20: showS

Fig. 2 caption: indicate the year and number of sites over the U.S., and note the small

differences between blue and green dashed lines.

Fig.3 caption: annual mean surface
* * *

---

## Author Comment (AC1) · 17 Jun 2019

We thank the referee for the detailed reviews and constructive comments that help to improve the manuscript. Below we respond to the comments in detail. (Referee's comments are in Italic). The manuscript has been revised accordingly.

*The authors aim to improve wet deposition simulation of nitric acid, nitrate and ammonium over the United States using GEOS-Chem via updating both in cloud stratiform cloud scavenging and below cloud washout. For in cloud scavenging, they adopt the*

[Figure]

*dynamic varied condensed water content provided by MERRA2 meteorological fields needed in wet scavenging parameterization, instead of the current assumption of a global flat value. For below cloud washout, they derive a new set of empirical washout scavenging coefficient and exponential coefficient for nitric acid based on the size-resolved coefficients summarizing from field measurement and theoretical derivation. This is an interesting and valuable study. The study would have a potential impact on the broad atmospheric composition study via improving tracers' wet scavenging if the authors could validate their work for other aerosols and their precursors. A minor revision is required before the paper is published in GMD.*

We appreciate the referee's positive comments about the importance of this study.

*Major Comments The authors test the physical-based condensed cloud water for stratiform cloud rainout. Convective cloud removal is important and is necessary to be studied as well. Studying convective rainout is particularly important for using the current generation of NASA GEOS meteorological fields since its partitioning of large scale and convective clouds tilts more towards the latter. The convective cloud fraction and water content can be provided by the GEOS model.*

This is a valid point. However, the convective cloud fraction and water content, while available in GEOS online simulation, is not available in GMAO reanalysis datasets (including MERRA2) used to drive GEOS-Chem. Therefore, as we have already pointed out in the last paragraph of Session 2.2, "the updated wet scavenging method discussed above for stratiform precipitation cannot be directly applied to convective precipitation rainout scavenging in GEOS-Chem". We agree with the referee that "Convective cloud removal is important and is necessary to be studied as well" and have pointed this out in the discussion session.

*The authors are highly encouraged to evaluate and summarize the impact of their work on other aerosols and their precursors. Once the GEOS-Chem adopts the improvements in wet scavenging parameterization suggested by the authors, all aerosols and*

*their precursors undergoing wet scavenging will be impacted. To have confidence in
using their work, they should at least provide a brief description of the model per-
formance for all important aerosol fields in supplementary material. In addition, the
authors' work focuses on the United States only. What is the anticipated influence of
the improved wet scavenging on other regions?*

Yes, we are evaluating the impacts of updated scheme on other aerosols and other
regions. Based on preliminary comparisons with relevant measurements we analyzed
so far, the updated wet scavenging parameterization also improves the model perfor-
mance over Europe and Asia. More in-depth analysis is being carried out and we are
preparing another paper on the impacts of updated wet scavenging parameterization
on global simulation of aerosols in GEOS-Chem. We have pointed this out in the dis-
cussion session.

*To be more useful of the proposed work on wet deposition, more words are needed
about the broader impact of the study on the whole atmospheric chemistry community.
Can other global chemistry models adopt their improvement? Is there anything that
other modelers should be cautioned of in adopting their work?*

The updates of rainout can be adopted by any atmospheric chemistry models which
assume constant cloud condensation water. The empirical washout can help to reduce
the overestimation of nitric acid gas shown in the work of Bian et al. (2017) by most
of atmospheric chemistry models. Corresponding discussions have been added in the
Summary and Discussion section.

*Specific comments 1. Page 3 line 26 equation 1: Should the k in exponential term differ
with the k in the denominator of coefficient? For my understanding, the k in exponential
term, which is the first-order rainout rate, is linked to specific tracer species. One the
other hand, the k in denominator represents the generic conversion rate of cloud water
to precipitation. Please double check this. Please also give units of these fields and
parameters in equation 1.*

[Figure]

We double-checked this. As shown in the work (session 1.1) of Jacob et al. (2000), the k in exponential term and the k in the denominator of coefficient are the same for soluble species. In GEOS-Chem, the model assumes the first-order rainout rate for water soluble aerosols and nitric acid gas equals the generic conversion rate of cloud water to precipitation. The units have been added.

*2. Page 7 lines 4-19: How about aerosols? Should the washout scavenging coefficients of aerosols be adjusted accordingly?*

We applied empirical washout rate from Laakso et al. (2003) for water soluble aerosols and washout rate from Feng (2007) for water insoluble aerosols. Corresponding information has been updated in the revised text.

*3. Page 8 line 7: Please add one more case study. Similar to case study 4 but empirical washout rate of HNO3 is applied only to large scale precipitation. This case, combined with case 4, will give us further information about the relative washout contribution from large scale and convective scale precipitations.*

As shown in Figure 1 below, the exclusion of wash out by convective precipitation in GEOS-Chem has negligible impacts on surface level HNO3, nitrate, and ammonium over the US. It is because convective precipitation is large over Tropics and small over middle and high latitude continent. Convective precipitation over the US is 10-100 times smaller than large-scale precipitation over there.

*4. Page 8 line 7: Do the authors present the work of empirical washout rates for aerosols? Section 2.2 seems only give discuss for HNO3. What are the new empirical washout rates for aerosols?*

In this study, empirical washout rate is from Laakso et al. (2003) for water soluble aerosols, while washout rate is from Feng (2007) for water insoluble aerosols. We modified corresponding sentences in the paper.

*5. Page 9 lines 1-2: The change range shown here (from 150% to 24%) includes not*

[Figure]

*only using empirical washout rate, but also changing cloud condensation water.*

Corrected. We changed the value from 150 % to 125 %.

———————————————

**[GMDD](https://doi.org)**
[Figure]

[Figure]

**Fig. 1.** Monthly variations of mean for year 2011 showing the comparison between nitric acid (a), nitrate (b), and ammonium (c) mass concentrations observed at ground-based sites (black) and simulated by GC12

---

## Author Comment (AC2) · 17 Jun 2019

We thank the referee for the detailed reviews and constructive comments that help to improve the manuscript. Below we respond to the comments in detail. (Referee's comments are in Italic).

*This paper presented a revised wet scavenging parameterization that considers the spatiotemporal variability of cloud liquid water content and an empirical washout(below-cloud scavenging) rate in the GEOS-Chem global chemical transport model. The au-*

[Figure]

*thors showed that the updated parameterization significantly improves simulated annual mean (and seasonal) mass concentrations of nitric acid, nitrate, and am-monium as compared with surface observations over the U.S. This is an important contribution to the improvement of GEOS-Chem. Minor revision is recommended be-fore publication on GMD.*

We appreciate the referee's positive comments about the importance of this study.

*Major comments: The impact of updated wet scavenging on model simulations was only assessed at the surface level and for nitric acid, nitrate, and ammonium over the U.S. It's not shown how the updated treatment of scavenging affects the global aerosol simulations, especially the vertical profiles and other aerosol species (e.g., sulfate). Consider discussing this in the Summary and Discussions section. Lead-210 aerosol tracer has been used to test wet deposition in GEOS-Chem (e.g., Liu et al., 2001), and this updated scavenging parameterization will need to be tested with (at least) lead-210 before it is incorporated into the standard version of the model.*

Agree. We added additional discussions on these issues in the summary and discussions section. More in-depth analysis is being carried out and we are preparing another paper on the impacts of updated wet scavenging parameterization on all major aerosol species over the whole globe.

*Page 5, equation 4: 1). "CW is grid-box mean cloud water content". What's the corresponding variable name in MERRA-2? Does it include both cloud liquid (QL) and ice (QI), or QL only? 2). It's not clear why the rain water term "Pr*DeltaT" is needed. There is no prognostic precipitation (no raining condensate) in MERRA-2 or GEOS-5. Prognostic cloud liquid and ice are autoconverted to estimate precipitation. Are "CW" values for pre-conversion or post-conversion? More explanation as well as references are needed.*

CW is "QL" in MERRA2. It only includes cloud liquid. As shown in Equation 6 in MERRA2's file specification (Bosilovich et al., 2016), QL is the residual condensation

water after precipitation. Due to large fraction of cloud water converted to rain water, cloud water in MERRA2 is low when precipitation is occurring. Because the fraction of soluble species rained out should equal to the fraction of total condensed water (or ICCW in our case) converted to rain water, we think that ICCW in Eq (3) should include rain water (i.e., Eq 4). The following reference is added to the reference list. Bosilovich, M. G., R. Lucchesi, and M. Suarez, 2016: MERRA-2: File Specification. GMAO Office Note No. 9 (Version 1.1), 73 pp, available from http://gmao.gsfc.nasa.gov/pubs/office_notes.

*Page 7, lines 20-23: The first-order rainout parameterization is not used for convective precipitation scavenging in GEOS-Chem driven by MERRA-2. Instead, scavenging in convective updrafts are coupled with convective transport (e.g., see section 2.3.1 of Liu et al., 2001).*

We have modified the text to reflect this.

*Minor comments: Title: Suggest adding "surface" to the title since this study examined the impact of revised scavenging on surface aerosol concentrations only.*

Accepted.

*Page 1, line 21: typo "mentoring" ("monitoring")*

Revised.

*Page 3, lines 6-7: are there references for this statement?*

We did the simulations with $4° \times 5°$ and $2° \times 2.5°$ horizontal resolutions in GEOS-Chem and found the switching of model resolution has small impact on simulated nitrate over the US.

*Page 3, line 14: change "in-site observations" to "surface observations"*

Accepted.

[Figure]

*Page 3: A brief description of the GEOS-Chem model is needed here before discussing the wet scavenging scheme.*

Accepted.

*Page 3, section 2: See this webpage http:// acmg.seas.harvard.edu/ geos/ geos_ chem_narrative.html for "Narrative description (and how to cite GEOS-Chem)", which provides guidance on citing relevant model components. "The wet deposition scheme in GEOS-Chem is described by Liu et al. [2001] for water-soluble aerosols and by Amos et al. [2012] for gases. Scavenging of aerosol by snow and cold/mixed precipitation is described by Wang et al. [2011, 2014]." Suggest citing Jacob et al. (2000) along with one of these publications, where appropriate, since it is an unpublished document. The first-order rainout parameterization (equations 1 and 2) is based on Giorgi and Chameides (1986), which also needs to be referenced.*

Added description and citation as suggested.

*Specify the units for variables in all equations in the text.*

Accepted.

*Page 4, line 3: condensed water content includes liquid and ice phases. Do you revise warm cloud scavenging only here? Does "Pr" (rate of new precipitation formation) include snow? How about ice cloud scavenging?*

We only revised rainout for warm cloud in this study. Pr, DQRLSAN in MERRA2, only includes rainwater. Ice cloud scavenging is applied to aerosols via washout by snow following the approach suggested by Wang et al. (2011).

*Page 4, lines 23-25: Croft et al. (2016) previously used GEOS-5 cloud liquid and ice water content to replace the fixed value in their GEOS-Chem-TOMAS simulations. Consider citing that work here. (Croft, B., Martin, R. V., Leaitch, W. R., Tunved, P., Breider, T. J., D'Andrea, S. D., and Pierce, J. R.: Processes controlling the annual cycle of Arctic aerosol number and size distributions, Atmos. Chem. Phys., 16, 3665-*

[Figure]

*3682, https://doi.org/10.5194/acp-16-3665-2016, 2016.)*

Thanks for pointing us to this work. The major difference of rainout treatment between Croft et al. (2016) and our work is the assumption of ICCW. Croft et al. (2016) used cloud liquid and ice water content to replace the fixed ICCW, while we used the sum of cloud liquid water and rain water to replace the fixed ICCW which is critical for rainout calculation (Eqs. 2-3). Corresponding discussions have been added in the revised paper.

*Page 8, line 4: these references are not for rainout and washout parameterizations, but for the standard GEOS-Chem model (or other model components).*

Accepted. These references are cited at the brief description of the GEOS-Chem model in revised paper.

*Page 9, line 13: CCW or ICCW?*

It is ICCW. Revised.

*Page 9, line 17: concentrations OF; line 20: showS*

Revised.

*Fig. 2 caption: indicate the year and number of sites over the U.S., and note the small differences between blue and green dashed lines.*

Modified as suggested.

*Fig.3 caption: annual mean surface*

Modified as suggested.

---

## Author Response (AR1)

We thank the referee for the detailed reviews and constructive comments that help to improve the manuscript. Below we respond to the comments in detail. (Referee's comments are in Italic). The manuscript has been revised accordingly.

The authors aim to improve wet deposition simulation of nitric acid, nitrate and ammonium over the United States using GEOS-Chem via updating both in cloud stratiform cloud scavenging and below cloud washout. For in cloud scavenging, they adopt the dynamic varied condensed water content provided by MERRA2 meteorological fields needed in wet scavenging parameterization, instead of the current assumption of a global flat value. For below cloud washout, they derive a new set of empirical washout scavenging coefficient and exponential coefficient for nitric acid based on the size-resolved coefficients summarizing from field measurement and theoretical derivation. This is an interesting and valuable study. The study would have a potential impact on the broad atmospheric composition study via improving tracers' wet scavenging if the authors could validate their work for other aerosols and their precursors. A minor revision is required before the paper is published in GMD.

We appreciate the referee's positive comments about the importance of this study.

**Major Comments**

The authors test the physical-based condensed cloud water for stratiform cloud rainout. Convective cloud removal is important and is necessary to be studied as well. Studying convective rainout is particularly important for using the current generation of NASA GEOS meteorological fields since its partitioning of large scale and convective clouds tilts more towards the latter. The convective cloud fraction and water content can be provided by the GEOS model.

This is a valid point. However, the convective cloud fraction and water content, while available in GEOS online simulation, is not available in GMAO reanalysis datasets (including MERRA2) used to drive GEOS-Chem. Therefore, as we have already pointed out in the last paragraph of Session 2.2, "the updated wet scavenging method discussed above for stratiform precipitation cannot be directly applied to convective precipitation rainout scavenging in GEOS-Chem". We agree with the referee that "Convective cloud removal is important and is necessary to be studied as well" and have pointed this out in the discussion session.

The authors are highly encouraged to evaluate and summarize the impact of their work on other aerosols and their precursors. Once the GEOS-Chem adopts the improvements in wet scavenging parameterization suggested by the authors, all aerosols and their precursors undergoing wet scavenging will be impacted. To have confidence in using their work, they should at least provide a brief description of the model performance for all important aerosol fields in supplementary material. In addition, the authors' work focuses on the United States only. What is the anticipated

**influence of the improved wet scavenging on other regions?**

Yes, we are evaluating the impacts of updated scheme on other aerosols and other regions. Based on preliminary comparisons with relevant measurements we analyzed so far, the updated wet scavenging parameterization also improves the model performance over Europe and Asia. More in-depth analysis is being carried out and we are preparing another paper on the impacts of updated wet scavenging parameterization on global simulation of aerosols in GEOS-Chem. We have pointed this out in the discussion session.

To be more useful of the proposed work on wet deposition, more words are needed about the broader impact of the study on the whole atmospheric chemistry community. Can other global chemistry models adopt their improvement? Is there anything that other modelers should be cautioned of in adopting their work?

The updates of rainout can be adopted by any atmospheric chemistry models which assume constant cloud condensation water. The empirical washout can help to reduce the overestimation of nitric acid gas shown in the work of Bian et al. (2017) by most of atmospheric chemistry models. Corresponding discussions have been added in the Summary and Discussion section.

**Specific comments**

1. Page 3 line 26 equation 1: Should the k in exponential term differ with the k in the denominator of coefficient? For my understanding, the k in exponential term, which is the first-order rainout rate, is linked to specific tracer species. One the other hand, the k in denominator represents the generic conversion rate of cloud water to precipitation. Please double check this. Please also give units of these fields and parameters in equation 1.

We double-checked this. As shown in the work (session 1.1) of Jacob et al. (2000), the k in exponential term and the k in the denominator of coefficient are the same for soluble species. In GEOS-Chem, the model assumes the first-order rainout rate for water soluble aerosols and nitric acid gas equals the generic conversion rate of cloud water to precipitation. The units have been added.

**2. Page 7 lines 4-19: How about aerosols? Should the washout scavenging coefficients of aerosols be adjusted accordingly?**

We applied empirical washout rate from Laakso et al. (2003) for water soluble aerosols and washout rate from Feng (2007) for water insoluble aerosols. Corresponding information has been updated in the revised text.

3. Page 8 line 7: Please add one more case study. Similar to case study 4 but empirical washout rate of HNO3 is applied only to large scale precipitation. This case,

combined with case 4, will give us further information about the relative washout contribution from large scale and convective scale precipitations.

As shown in Figure S1 below, the exclusion of wash out by convective precipitation in GEOS-Chem has negligible impacts on surface level HNO3, nitrate, and ammonium over the US. It is because convective precipitation is large over Tropics and small over middle and high latitude continent. Convective precipitation over the US is 10-100 times smaller than large-scale precipitation over there.

Figure S1. Monthly variations of mean for year 2011 showing the comparison between nitric acid (a), nitrate (b), and ammonium (c) mass concentrations observed at ground-based sites (black) and simulated by GC12 (blue), Knew (yellow dash), ICCWt (green), ICCWt EW, and ICCWt EWL (blue circular points) cases.

4. Page 8 line 7: Do the authors present the work of empirical washout rates for aerosols? Section 2.2 seems only give discuss for HNO3. What are the new empirical washout rates for aerosols?

In this study, empirical washout rate is from Laakso et al. (2003) for water soluble aerosols, while washout rate is from Feng (2007) for water insoluble aerosols. We modified corresponding sentences in the paper.

5. Page 9 lines 1-2: The change range shown here (from 150% to 24%) includes not only using empirical washout rate, but also changing cloud condensation water.

Corrected. We changed the value from 150 % to 125 %.

We thank the referee for the detailed reviews and constructive comments that help to improve the manuscript. Below we respond to the comments in detail. (Referee's comments are in Italic).

This paper presented a revised wet scavenging parameterization that considers the liquid spatiotemporal variability of cloud water content and an empirical washout(below-cloud scavenging) rate in the GEOS-Chem global chemical transport model. The authors showed that the updated parameterization significantly improves simulated annual mean (and seasonal) mass concentrations of nitric acid, nitrate, and am-monium as compared with surface observations This is an important contribution to the improvement of over the U.S.GEOS-Chem. Minor revision is recommended be-fore publication on GMD.

We appreciate the referee's positive comments about the importance of this study.

**Major comments:**

The impact of updated wet scavenging on model simulations was only assessed at the surface level and for nitric acid, nitrate, and ammonium over the U.S. It's not shown how the updated treatment of scavenging affects the global aerosol simulations, especially the vertical profiles and other aerosol species (e.g., sulfate). Consider discussing this in the Summary and Discussions section. Lead-210 aerosol tracer has been used to test wet deposition in GEOS-Chem (e.g., Liu et al., 2001), and this updated scavenging parameterization will need to be tested with (at least) lead-210 before it is incorporated into the standard version of the model.

Agree. We added additional discussions on these issues in the summary and discussions section. More in-depth analysis is being carried out and we are preparing another paper on the impacts of updated wet scavenging parameterization on all major aerosol species over the whole globe.

Page 5, equation 4: 1). "CW is grid-box mean cloud water content". What's the corresponding variable name in MERRA-2? Does it include both cloud liquid (QL) and ice (QI), or QL only? 2). It's not clear why the rain water term "Pr\*DeltaT" is needed. There is no prognostic precipitation (no raining condensate) in MERRA-2 or GEOS-5. Prognostic cloud liquid and ice are autoconverted to estimate precipitation. Are "CW" values for pre-conversion or post-conversion? More explanation as well as references are needed.

CW is "QL" in MERRA2. It only includes cloud liquid. As shown in Equation 6 in MERRA2's file specification (Bosilovich et al., 2016), QL is the residual condensation water after precipitation. Due to large fraction of cloud water converted to rain water, cloud water in MERRA2 is low when precipitation is occurring. Because the fraction of soluble species rained out should equal to the fraction of total condensed water (or ICCW in our case) converted to rain water, we think that ICCW in Eq (3) should include rain water (i.e., Eq 4). The following reference is added to the reference list.

Bosilovich, M. G., R. Lucchesi, and M. Suarez, 2016: MERRA-2: File Specification. GMAO Office Note No. 9 (Version 1.1), 73 pp, available from http://gmao.gsfc.nasa.gov/pubs/office\_notes.

Page 7, lines 20-23: The first-order rainout parameterization is not used for convective precipitation scavenging in GEOS-Chem driven by MERRA-2. Instead, scavenging in convective updrafts are coupled with convective transport (e.g., see section 2.3.1 of Liu et al., 2001).

We have modified the text to reflect this.

Minor comments:

Title: Suggest adding "surface" to the title since this study examined the impact of revised scavenging on surface aerosol concentrations only.

Accepted.

Page 1, line 21: typo "mentoring" ("monitoring")

Revised.

Page 3, lines 6-7: are there references for this statement?

We did the simulations with  $4^{\circ} \times 5^{\circ}$  and  $2^{\circ} \times 2.5^{\circ}$  horizontal resolutions in GEOS-Chem and found the switching of model resolution has small impact on simulated nitrate over the US.

Page 3, line 14: change "in-site observations" to "surface observations"

Accepted.

Page 3: A brief description of the GEOS-Chem model is needed here before discussing the wet scavenging scheme.

Accepted.

Page3,section2:Seethiswebpagehttp://acmg.seas.harvard.edu/geos/geos\_chem\_narrative.htmlfor"Narrativedescription(and how to cite GEOS-Chem)", which provides guidance on citingrelevant model components."The wet deposition scheme in GEOS-Chem is describedby Liu et al.[2001] for water-soluble aerosols and by Amos et al.[2012] for gases.Scavenging of aerosol by snow and cold/mixed precipitation is described by Wang et

al. [2011, 2014]." Suggest citing Jacob et al. (2000) along with one of these publications, where appropriate, since it is an unpublished document. The first-order rainout parameterization (equations 1 and 2) is based on Giorgi and Chameides (1986), which also needs to be referenced.

Added description and citation as suggested.

Specify the units for variables in all equations in the text.

Accepted.

Page 4, line 3: condensed water content includes liquid and ice phases. Do you revise warm cloud scavenging only here? Does "Pr" (rate of new precipitation formation) include snow? How about ice cloud scavenging?

We only revised rainout for warm cloud in this study. Pr, DQRLSAN in MERRA2, only includes rainwater. Ice cloud scavenging is applied to aerosols via washout by snow following the approach suggested by Wang et al. (2011).

Page 4, lines 23-25: Croft et al. (2016) previously used GEOS-5 cloud liquid and ice water content to replace the fixed value in their GEOS-Chem-TOMAS simulations. Consider citing that work here. (Croft, B., Martin, R. V., Leaitch, W. R., Tunved, P., Breider, T. J., D'Andrea, S. D., and Pierce, J. R.: Processes controlling the annual cycle of Arctic aerosol number and size distributions, Atmos. Chem. Phys., 16, 3665-3682, https://doi.org/10.5194/acp-16-3665-2016, 2016.)

Thanks for pointing us to this work. The major difference of rainout treatment between Croft et al. (2016) and our work is the assumption of ICCW. Croft et al. (2016) used cloud liquid and ice water content to replace the fixed ICCW, while we used the sum of cloud liquid water and rain water to replace the fixed ICCW which is critical for rainout calculation (Eqs. 2-3). Corresponding discussions have been added in the revised paper.

*Page 8, line 4: these references are not for rainout and washout parameterizations, but for the standard GEOS-Chem model (or other model components).*

Accepted. These references are cited at the brief description of the GEOS-Chem model in revised paper.

Page 9, line 13: CCW or ICCW?

It is ICCW. Revised.

Page 9, line 17: concentrations OF; line 20: showS

Revised.

Fig. 2 caption: indicate the year and number of sites over the U.S., and note the small differences between blue and green dashed lines.

Modified as suggested.

Fig.3 caption: annual mean surface

Modified as suggested.

[revised manuscript text omitted]
., 17,</li>                                                                                                                                                                                                                                                                                                                                                                                                                                                |  |  |  |  |  |  |  |
|  <li>Blah, H., Chin, M., Haugidstahle, D. A., Schulz, M., Myne, G., Bauel, S. E., Eund, M. T.,</li> <li>Karydis, V. A., Kucsera, T. L., Pan, X., Pozzer, A., Skeie, R. B., Steenrod, S. D.,</li> <li>Sudo, K., Tsigaridis, K., Tsimpidi, A. P., and Tsyro, S. G., Investigation of global</li> <li>particulate nitrate from the AeroCom phase III experiment, Atmos. Chem. Phys., 17,</li> <li>12911-12940, https://doi.org/10.5194/acp-17-12911-2017, 2017.</li>                                                                                                                                                                                                                                                                      |  |  |  |  |  |  |  |
|  <li>Blah, H., Chin, M., Haugidstahle, D. A., Schulz, M., Mynie, G., Bauel, S. E., Eund, M. T., Karydis, V. A., Kucsera, T. L., Pan, X., Pozzer, A., Skeie, R. B., Steenrod, S. D., Sudo, K., Tsigaridis, K., Tsimpidi, A. P., and Tsyro, S. G., Investigation of global particulate nitrate from the AeroCom phase III experiment, Atmos. Chem. Phys., 17, 12911-12940, https://doi.org/10.5194/acp-17-12911-2017, 2017.</li> <li>Bosilovich, M. G., R. Lucchesi, and M. Suarez: MERRA-2: File Specification. GMAO</li>                                                                                                                                                                                                        |  |  |  |  |  |  |  |
|  <li>Blah, H., Chin, M., Haugidstahle, D. A., Schulz, M., Myne, G., Bauel, S. E., Eund, M. T., Karydis, V. A., Kucsera, T. L., Pan, X., Pozzer, A., Skeie, R. B., Steenrod, S. D., Sudo, K., Tsigaridis, K., Tsimpidi, A. P., and Tsyro, S. G., Investigation of global particulate nitrate from the AeroCom phase III experiment, Atmos. Chem. Phys., 17, 12911-12940, https://doi.org/10.5194/acp-17-12911-2017, 2017.</li> <li>Bosilovich, M. G., R. Lucchesi, and M. Suarez: MERRA-2: File Specification. GMAO Office Note No. 9 (Version 1.1), 73 pp, available from</li>                                                                                                                                                         |  |  |  |  |  |  |  |
|  <li>Blah, H., Chin, M., Haugidstahle, D. A., Schulz, M., Wyne, G., Badel, S. E., Eund, M. L., Karydis, V. A., Kucsera, T. L., Pan, X., Pozzer, A., Skeie, R. B., Steenrod, S. D., Sudo, K., Tsigaridis, K., Tsimpidi, A. P., and Tsyro, S. G., Investigation of global particulate nitrate from the AeroCom phase III experiment, Atmos. Chem. Phys., 17, 12911-12940, https://doi.org/10.5194/acp-17-12911-2017, 2017.</li> <li>Bosilovich, M. G., R. Lucchesi, and M. Suarez: MERRA-2: File Specification. GMAO Office Note No. 9 (Version 1.1), 73 pp, available from http://gmao.gsfc.nasa.gov/pubs/office_notes, 2016.</li>                                                                                                      |  |  |  |  |  |  |  |
|  <li>Bian, H., Chin, M., Hadgustaine, D. A., Schulz, W., Mynre, O., Badel, S. E., Edid, W. T., Karydis, V. A., Kucsera, T. L., Pan, X., Pozzer, A., Skeie, R. B., Steenrod, S. D., Sudo, K., Tsigaridis, K., Tsimpidi, A. P., and Tsyro, S. G., Investigation of global particulate nitrate from the AeroCom phase III experiment, Atmos. Chem. Phys., 17, 12911-12940, https://doi.org/10.5194/acp-17-12911-2017, 2017.</li> <li>Bosilovich, M. G., R. Lucchesi, and M. Suarez: MERRA-2: File Specification. GMAO Office Note No. 9 (Version 1.1), 73 pp, available from http://gmao.gsfc.nasa.gov/pubs/office_notes, 2016.</li> <li>Croft, B., Martin, R. V., Leaitch, W. R., Tunved, P., Breider, T. J., D'Andrea, S. D., and</li>  |  |  |  |  |  |  |  |
|                                                                                                                                                                                                                                                                                                                                                                                                                                                                                                                                                                                                                                                                                                                                                 |  |  |  |  |  |  |  |

[revised manuscript text omitted]

simulated annual mean values for the 3 species by GC12, Knew, ICCWt, and ICCWt EW 4 5 cases.

|                          |                  | HNO 3 |      |                       | NIT   |      |                       | NH4  |      |
|--------------------------|------------------|------------------|------|-----------------------|-------|------|-----------------------|------|------|
|                          | Mean             | NMB              | r    | Mean                  | NMB   | r    | Mean                  | NMB  | r    |
|                          | $(\mu g m^{-3})$ | (%)              |      | (µg m -3 ) | (%)   |      | (µg m -3 ) | (%)  |      |
| Observation              | 0.83             |                  |      | 0.70                  |       |      | 0.60                  |      |      |
| GC12                     | 2.04             | 145.1            | 0.73 | 1.89                  | 168.1 | 0.53 | 1.09                  | 81.4 | 0.75 |
| Knew                     | 2.05             | 146.8            | 0.73 | 1.90                  | 170.5 | 0.53 | 1.11                  | 84.5 | 0.75 |
| ICCW t | 1.87             | 125.0            | 0.74 | 1.29                  | 83.5  | 0.57 | 0.86                  | 42.7 | 0.78 |
| ICCW t _EW    | 1.03             | 24.2             | 0.72 | 0.88                  | 25.0  | 0.57 | 0.68                  | 12.8 | 0.78 |